# Heat Shock Protein and Disaggregase Influencing the Casein Structuralisation

**DOI:** 10.3390/ijms26136360

**Published:** 2025-07-01

**Authors:** Irena Roterman, Katarzyna Stapor, Dawid Dułak, Leszek Konieczny

**Affiliations:** 1Department of Bioinformatics and Telemedicine, Jagiellonian University—Medical College, Medyczna 7, 30-688 Krakow, Poland; 2Department of Applied Informatics, Faculty of Automatic, Electronics and Computer Science, Silesian University of Technology, Akademicka 16, 44-100 Gliwice, Poland; katarzyna.stapor@polsl.pl; 3ABB Business Services Sp. z. o. o. ul., Zagańska 1, 04-713 Warszawa, Poland; dawid.dulak@gmail.com; 4Medical Biochemistry, Jagiellonian University—Medical College, Kopernika 7, 31-034 Krakow, Poland; mbkoniec@cyf-kr.edu.pl

**Keywords:** chaperonin, disaggregase, casein, protein folding, Hsp104, hydrophobicity

## Abstract

The contribution of the environment to protein folding seems obvious. The aqueous environment directs protein folding towards generating a centric hydrophobic core with a polar shell. The cell membrane environment—in which numerous proteins are anchored—to stabilise the arrangement, expects the exposure of hydrophobic residues and the concentration of polar residues in the central part—a channel for the transport of numerous molecules. The influence of these environments seems evident due to the persistent residence of proteins in their surroundings providing an external force field for structure stabilisation. Structural forms are also obtained with the participation of supporting proteins—such as proteins from the heat shock protein group—which accompany the folding process and temporarily provide an appropriate external force field in which the protein, having obtained the correct structure for its activity, is released from interaction with the supporting protein. This paper discusses an example of the contribution of Hsp104 to casein folding and the effect of disaggregase preventing inappropriate aggregation. For this purpose, a model called the fuzzy oil drop (FOD-M) was used, which takes hydrophobic interactions into account in the assessment of protein structure status. Their distribution in the protein body highlights the contribution and influence of the external force field—originating from Hsp104 and the disaggregase in this case.

## 1. Introduction

Proteins referred to as heat shock proteins are proteins that are intensively produced when the temperature of the environment of an organism is elevated [1,2,3,4]. The presence of these proteins, referred to as supporting proteins, prevents obtaining misfolded structures that do not exhibit biological activity. The promotion of the folding process also takes place in other situations that deviate from the standard environmental conditions of the cell, including those associated with mechanical stress [5]. The importance of the Hsp activity is observed for the tumourigenesis processes. Therapeutic techniques based on exploiting the activity of these proteins are being sought [6,7]. The Hsp activity is also used in diagnostics and therapy related to immunological techniques [8,9,10]. Hopes for the appropriate use of the Hsp proteins have also been linked to the treatment of neurodegenerative diseases [11]. A significant proportion of the Hsp proteins are chaperones and chaperonins [12,13,14,15]. Just like the folding process—the adoption of a structure with coded biological activity—the aggregation process is also controlled. Preventing the generation of inappropriate complexes becomes important in the context of neurodegenerative diseases, where the uncontrolled formation of complexes in the form of fibrils of unlimited length poses a threat to the functioning of the entire organism [16].

The importance and role of Hsp in the folding process and the role of disaggregase in preventing the formation of an abnormal complex are the objects of this paper. The presence of the proteins supporting these two processes is regarded as providing an external force field directing the process in question by adapting the structure to the conditions of the field. A similar analysis discusses the role of supporting proteins participating in the folding process: prefoldin [17], chaperon [18], and chaperonin [19].

The folding of proteins in an aqueous environment obviously leads to micelle-like structures with a centric hydrophobic core and a polar surface (Figure 1A). Such a model relies on the basic phenomenon of micelle formation by bipolar molecules. Treating amino acids as a collection of 20 bipolar molecules with a varying ratio of polar to hydrophobic parts, it can be assumed that the polypeptide chain in an aqueous environment directs structuring towards the exposure of polar residues on the surface with the centralisation of hydrophobic residues. Such a mechanism leads to the presence of a hydrophobic core in numerous proteins active in aqueous environments. A different distribution of amino acids is expected in membrane-anchored proteins, where the exposure of hydrophobic residues is expected for stabilisation in this environment (Figure 1B).

The analysis of the Hsp104 protein in a complex with a casein chain being folded aims to quantify the contribution of chaperonin to the structuring of the polypeptide. The analysis was based on a model called the fuzzy oil drop model (FOD-M), which uses a form of hydrophobicity/polarity distribution within the protein body to assess and characterise the protein structure [17,18,19].

The structure of the chaperonin itself as well as the protein being folded was assessed. A mechanism has also been proposed for obtaining a chaperonin structure that shows a structuring far from what was expected in the aqueous environment. This raises the question of how it is possible to generate—in an aqueous environment—a structure that is far from what that environment prefers [20].

In addition, the results of the analysis of the disaggregase complex with the casein polypeptide are presented, determining the status of the two external force field providers in both processes.

## 2. Results

The FOD-M model classifies protein structures using two parameters: RD and K. The former determines the degree to which the hydrophobicity distribution in the protein body approximates a micelle-like idealised arrangement with a centric hydrophobic core and a polar surface. In consequence the larger the Rd value the larger the discordance of hydrophobicity distribution with respect to the idealised one. The latter parameter assesses the degree to which non-aqueous factors are involved in the folding process and therefore the source of the mismatch to the micelle-like distribution. The contribution of these factors that change the specificity of the aqueous environment provides the folding protein with a kind of matrix to which the polypeptide chain will adapt its structuring.

### 2.1. Structure of the Hsp104 Chaperonin Controlling the Casein Folding Process

The structure of chaperonin Hsp104 is composed of six chains to form a six-homo complex. The complete structure of the entire complex including the caseine chain folded in the chaperonin environment was assessed. The structure itself without the “guest” molecule was also assessed, as well as the status of the individual chains. In the single chain structure, the PDBSum database [21] does not provide a division into domains according to CATH [22]. However, as the individual parts of the chain form distinct independent subunits, so-called pseudo-domains were introduced guided by the local packing of the individual parts of the chain.

A summary of the results is given in Table 1.

The structure of the complex form of chaperonin is given in Figure 2. where the presence of the “guest” casein chain is also indicated.

The assessment of the structuring of the complex CHAINS A-F—the chaperonin +P “guest” molecule of the protein being folded—in this case, a fragment of the casein chain, shows the ordering of the hydrophobicity distribution far from that of the idealised T distribution. This complex has no hydrophobic core in its structure. A very high value of the K parameter indicates an ordering not adapted to the orientation resulting from the polar environment.

The structure of chaperonin Hsp104 itself shows a very similar degree of disordered hydrophobicity distribution expressed with relatively high *RD* as well as *K* values.

The corresponding distributions expressing the *T*, *O*, and *M* profile for *K* = 2.1 are given in Figure 3A. Clearly the *O* distribution shows comparable levels throughout the chains. This can be interpreted as an *R*-type distribution. This implies a significant “distance” from the arrangement with a centric hydrophobic core (blue line—Figure 3). It also means isolation from the conditions imposed by the polar environment. The Hsp104 molecule thus provides a distinct external force field for the folding polypeptide, based on a uniform distribution of hydrophobicity.

An interpretation of the biological activity of chaperonin Hsp104 based on the criteria of the FOD-M model identifies the status of the chaperonin (chains A-F) as highly dissimilar to an idealised micelle-like arrangement with a centric hydrophobic core and a polar surface. The value of *K* = 2.1 indicates a significant deviation from the idealised state. This condition is caused by the form of the O distribution, which more closely approximates the uniform R distribution. The Oi values for the individual residues oscillate around an almost straight-line distribution. The absence of a hydrophobic core is significant. The expected high Ti values (Figure 3) are not represented in the protein body. On the contrary, a significant deficit in hydrophobicity is evident in these locations. The graph O in Figure 3A illustrates the mismatch status of this distribution with the expected Ti. The Oi chart, however, has additional significance. This diagram expresses the type of force field under which the casein chain folds. The Oi diagram is an expression of the form of the external force field providing the environment for the folding protein. A straight-line course for this field means complete isolation from the aqueous environment eliminating its influence on the formation of the casein chain being folded.

Chains treated as individual structural units (3D Gaussian function constructed for each chain individually—Table 1) show comparable values of R and K parameters. The RD parameter indicates an ordering of hydrophobicity far from the T distribution, while the values of the K parameters are already much lower in relation to the analogous K value for the complex (Table 1, Figure 3 and Figure 4).

The status of the casein chain is assessed in approximation. The source file 5VJH does not provide the casein chain sequence. The coordinates are provided for this chain with the exact positions of the backbone atoms and the positions of the atoms referred to as CBeta. Therefore, this chain was given the form of poly-alanine. The status of this chain is not significantly affected due to the low probability of interaction of neighbouring residues in a highly Beta-structured side-chain orientation arrangement. The status of the chain thus developed (Figure 5) reveals a level of hydrophobicity oscillating only around a constant level throughout the chain.

The native structural form of the casein chain in the aqueous environment is not available due to the specificity of this protein appearing in an aqueous environment in the form of micelles. Theoretically, the aqueous environment would not accept such an extended form of the chain, imposing a globular form. From the point of view of both enthalpy and entropy, the extended form of the polypeptide chain with a length of 26 aa is not possible as stable.

The status of the chain being folded is determined by the very high value of the RD parameter and the value K = 1.0 (Figure 5).

In the structure of a single chain of Hsp194, four pseudo-domains were distinguished (the presence of domains according to CATH is not indicated) as shown in Figure 6. Pseudo-domains were identified based on the clear variation in the packing degree of the individual chain fragments. The chain fragments distinguished as pseudo-domains are shown in Figure 3B.

The pseudo-domains are the following: PD1 (165–340)—yellow 0–175, PD2 (344–557)—green, PD3 (176–265), PD3 (558–775)—pink, PD4 (266–483), and PD5 (776–884)—red.

The question arises: How is it possible to generate a chaperonin structure that is so different from the specifics of the aqueous environment?

For this purpose, individual pseudo-domains were analysed by treating them as individual structural units (Figure 6—the division is also shown in Figure 3B). A proper 3D Gaussian function was generated for each of these units, and the status of each pseudo-domain was determined. Their characteristics are given in Table 1. Interpretation of the results regarding the status of the individual pseudo-domains indicates that they are constructed highly in line with expectations and orientation by the water environment. Very low K values indicate the influence of the polar environment. This is illustrated by the sets of the T, O, and M profiles (for the respective K values given in the legend), where a high correspondence between the T and O distributions is evident (Figure 7 and Figure 8).

The essence of achieving complete chaperonin structuring lies in the appropriate interaction of the individual chains. This is given in Figure 9 and Figure 10. The status of residues involved in interchain interactions was demonstrated as engaging the hydrophobic interactions of hydrophobic residues exposed on the surface of domains (the Oi level higher than Ti).

The juxtaposition of the T and O profiles for the residues involved in the interchain interaction reveals the use of local excess of hydrophobicity (first half of the T and O profiles—Figure 9A) and the involvement of polar residues showing a deficit in hydrophobicity (second part of the profiles—Figure 9A). Hydrophobicity deficits are often associated with the presence of a cavity. The geometric match to the corresponding cavity creates the conditions for stabilising the interchain interaction.

Similarly, the interaction of chain A with chain F shows the significant involvement of residues showing a local excess of hydrophobicity. This means that the contribution of this type of interaction plays a role in stabilising the complex.

The interaction of the A and B chains involves amino acids exclusively from the PD1 and PD3 domains (Figure 3A). In contrast, the interaction of chain A with chain F involves residues from all domains.

Based on this disproportion of the involvement of individual domains in the stabilisation of the complex, it can be concluded that the PD2 and PD4 domains (with much less involvement in interchain interactions) are the main providers of the external force field for creating the environment that determines the conditions for the folding process—the imposition of the structural form of chain P.

In contrast, the local exposure of hydrophobicity within individual domains (profiles shown in Figure 9) reveals the use of a local excess of hydrophobicity to generate a stable interchain arrangement.

Hence, it can be concluded that, despite the low K values for the individual pseudo-domains, local deficits and the presence of a local excess of hydrophobicity on the domain surface result in the emergence of an interchain interaction leading to an arrangement far from the idealised arrangement typical of an aqueous environment. Interactions of other types (beyond hydrophobic) are also present. However, the importance of hydrophobic interactions is proving to be significant in the aqueous environment surrounding the chaperonin (also shown in the 3D representation—Figure 10).

### 2.2. Mitochondrial Disaggregase from Homo Sapiens Preventing Incorrect Casein Complexation

The structure of the _PARL_Skd3 chaperone classified as disaggregase in a complex with casein peptide is available as PDB ID 7TTR [23].

This chaperonin takes the form of a spiral hexameric arrangement with the substrate (14-peptide of casein) located in the central part. This chaperonin prevents protein aggregation in the intermembrane space of the mitochondrion. In the structure in question, the substrate is a 14-peptide casein of undetermined sequence (Beta-casein from Bos taurus). As in the previous case, the alanine polypeptide was inserted, as the coordinates were only rendered with the accuracy of the C-Beta position.

The characteristics of the chaperonin in question given in Table 2 reveal its status from the point of view of organising the hydrophobicity/polarity relationship.

An interpretation based on the assumptions of the FOD-M model indicates that the external force field for the disaggregase in question is expressed as K = 1.6. The juxtaposition of the T, O, and M profiles for K = 1.6 reveals an R-type distribution—similar to the status for the chaperonin discussed above (PDB ID—5VJH).

The status of the complete complex as shown in Figure 11 reveals a high discordance taking the FOD-M-based criteria. The profiles shown in Figure 11A can be interpreted as the form of an external force field for casein, the status of which is also far from the expected result with respect to the micelle-like organisation (Figure 11C).

The status of the individual chain described by K = 0.7 visualises the approach to micelle-like organisation; however, its folding can not be treated as water directed. The analysis of the status of pseudo-domains, however, reveals the possible folding of individual domains as directed by an aqueous environment (Figure 12).

The characteristics of the disagreggase complex are similar to the previously discussed Hsp104. The external force field form with high RD and K values indicates isolation from the influence of the aqueous environment. The single chain status shows a structuring far from the micelle-like structure expected in the aqueous environment. The status of the individual domains, however, appears to be very close to being determined by the influence of the aqueous environment. The folding of a single chain in the form of a domain structure is achievable in the aqueous environment. Despite the very low K values, single amino acids or short fragments showing local hydrophobicity exposure are present in each chain (Figure 13) involved in an interaction with other chains of the same structure. This results in a structure far from the conditions created by the presence of the aqueous environment. Nevertheless, the use of local exposures dictates the appropriate structure of the complex.

The “protected” status of the casein chain (chain P in PDB ID—7TTR)—similarly to the Hsp104 example discussed above—is characterised by high RD and K values, implying the maintenance of a structuring far from that expected in the aqueous environment.

### 2.3. Function-Related Dynamic Structures of Disaggregase

The three structures of *Saccharomyces cerevisiae* disaggregase available from the PDB database represent three function-related forms: closed, extended, and open. The analysis of these structural forms given below reveals the roles of individual chains in generating function-related structural changes associated with the translocation of unfolded polypeptides.

The activity of the disaggregase in question is considered to be exceptional due to the rapid hydrolysis of ATP. The discussed three forms represent the catalytically inactive Hsp104 variant (Hsp104_DWB_) in the ATP-bound state adopting distinct ring conformations (closed, extended, and open). These forms are detected despite being in the same nucleotide state as it is declared in [24].

The results given in Table 3 suggest a high structural similarity measured by the status of the hydrophobicity distribution of the closed and extended form with the above-mentioned example of human disaggregase in complex with casein, both the status of individual chains as components of complexes (common 3D Gauss function for the complex) and the status of chains treated as individual structural units (3D Gauss function generated for each chain individually).

Clear differences are visible in the open form, where the status of the complex and individual chains, both treated as components of the complex and as individual structural units, changes significantly (Figure 14). The greatest changes were identified in chains B, C, and E as individual structural units. On the other hand, the greatest disturbance in the order of hydrophobicity distribution (compared to that observed in the closed form) is shown by chain F (Figure 15).

All other chains also show increased values of RD and K parameters, although chain F introduces a significant disturbance in the distribution of hydrophobicity. This differentiation of the status of individual chains within the complex proves a strong disturbance of the symmetry system, which is observed in other forms of the discussed complex, as well as in human disaggregase discussed above in the complex with casein. The significantly increased RD value for the entire complex in the open form proves the above-proposed interpretation.

In order to show the changes between the individual forms of the complex, a visualization using T, O, and M profiles is presented for the complexes discussed and for the chains showing the highest changes of status: Chain F is marked in all structural forms.

In the case of the discussed disaggregase, its important feature is the rapid hydrolysis of ATP in contrast to other Hsp104, which do not do this [24]. The structural changes analysed here are related to possible translocations of unfolded polypeptides.

The comparative analysis carried out for the discussed disaggregase structures in different forms related to the performed function shows the validity of the assessment based on the FOD-M model. The comparative analysis of the results describing disaggregase in its different states related to the biological function with the above-presented disaggregase in interaction with casein shows a consistent picture of Hsp104 in the interpretation based on the FOD-M model. The comparative analysis shows the similarity of the closed form with the discussed Hsp104 associated with casein. The dynamic forms of disaggregase differ slightly from the status of the static form (complex with casein), which indicates the possibility of assessing structural changes related to the biological function.

## 3. Discussion

This paper discusses an example of the action of the Hsp104 chaperonin as a protein that assists in the process of polypeptide chain folding by providing an external force field to which the polypeptide chain being folded adapts. An analysis of the role of the analogous proteins supporting the folding process—prefoldin [17], chaperone [18], and chaperonin [19]—indicates a similar mechanism for the folding process in the presence of chaperone proteins.

The conclusion drawn from the example presented concerns the need to take into account the dependence of the folding process on two functions: internal force field (non-binding interactions: electrostatic, vdW, and torsional potential) and external force field, expressed by the function shown in Equation (6) for a specific value of the K parameter. The problem of structure prediction on the basis of the reconstruction of the folding process belongs to the multiple minima problem category. The conformational space for a polypeptide with 100 aa takes the form of a large number of local minima. This large number of local minima turns out to be the result of the great potential for interactions at the atom–atom level (internal force field). It is also the result of the huge number of possibilities for environmental forms. Different environmental conditions are used by evolution for folding processes, such as folding in the aqueous environment, membranes, endoplasmic reticulum, Golgi apparatus, chaperones, and chaperonins. The evolution elaborated the use of environmental variation for folding processes.

The impact of the presence of proteins that support the folding process in terms of the FOD-M model has been discussed with the use of chaperone, chaperonin, and prefoldin [21,22,25] preventing the adoption of structuring before the protein being folded is delivered to the chaperonin and therefore to an environment that ensures the correct structure providing biological activity. Based on these observations, a modified form of the funnel model was proposed [26]. The introduction of a horizontal axis expressing the values of the K parameter—and thus the environmental variability—explains the orientation of the folding process towards a specific energy minimum [26].

Structural analysis of amyloid proteins based on the application of the FOD-M model leads to the determination of the degree of environmental change leading to amyloidogenesis. Three distinct scenarios have been proposed for this process [27]. In consequence, the definition of the funnel model to express the favouring of the formation of the corresponding amyloid form is possible [27].

Chaperonin provides an external force field directing the folding process by imposing a structural form to match the environment, which in this case is Hsp104. A consequence of this observation is the proposal to use the front Pareto method for simulating the in silico folding process [26,27]. In the front Pareto model, the function being optimised is a function of two functions. In the FOD-M model, these are internal force field and external force field. Optimisation of only one function leads to a disruption of the conditions of the other function. The two functions are in a contradictory relationship. Therefore, the simulation of the search for the functional structure of a protein involves finding a consensus between these two functions.

The role of the ordering of the hydrophobicity distribution in the final protein structures has been demonstrated in the analysis of so-called chameleon proteins, where a fragment with an identical sequence (8 aa) adopts different secondary forms in two different proteins: helical and beta-structural. The local status (hydrophobicity in the protein body) of these chain segments has been shown to be comparable from the point of view of hydrophobicity distribution irrespective of its secondary structure [28,29].

The need to take into account the external force field for protein structure prediction was also demonstrated based on the CASP results [30,31].

The structure of caseins is not known beyond the identification of nanomicelles, which, present in milk, play a variety of roles [32]. The formation of appropriate complexes in this context is proving critical to maintaining the correct biological activity of this protein. Hence, controlling both the folding process and preventing the formation of inappropriate complexes is critical for this protein.

Comparative analysis of Hsp104 forms in complex with the polypeptide chain of the folding protein and in biologically active forms suggests the validity of the assessment of the T, O, and M hydrophobicity distributions, which reveal the type of structural changes. They also reveal the importance and influence of the immediate environment on the folding protein by providing an external force field to prevent the process of adopting an incorrect structural form or forming incorrect complexes, as is seen in the case of caseins. The analysis of structures representing misfolded forms given in the Appendix A reveals the importance and role of the appropriate distribution of hydrophobicity in obtaining the correct structure for a given amino acid sequence.

## 4. Materials and Methods

### 4.1. Data

The object of the analysis is a complex of heat shock proteins—Hsp104 in a complex with casein. The structure available in the Protein Data Bank (PDB) [33]—ID 5VJH [34]—is analysed in this paper using the FOD-M model [17]. The aim of the analysis is to demonstrate the form of orienting the casein folding process in the external force field environment derived from Hsp104.

In addition, the specificity of the external force field of disaggregase for casein stabilising the structure of this protein preventing its complexation in an aqueous environment (PDB ID—7TTR [23]) was also analysed.

The application of the FOD-M model enables the assessment of the dynamics of structural changes related to the disaggregase function. The analysis of disaggregase forms, such as closed (PDB ID—6N8T [24]), open (PDB ID—6N8V [24]), and extended (PDB ID—6N8Z [24]), illustrates the type of changes and determines the roles of individual disaggregase components in the process of complexation of polypeptide chains, preventing their improper aggregation.

In order to justify the use of the FOD-M model as a tool expressing the characteristics of protein structuring, a numerical group of misfolded proteins was analysed. The provider of the set of such structural forms is the CASP project [35,36], where the biologically active structure is known and structural forms proposed as models are available, the degree of which is quantitatively determined with the GDT_TS scale used in this project. For this purpose, the target T1266_1-D1 from the CASP project16 was randomly selected. The analysis of the set of models representing misfolded forms is given in the Appendix A.

### 4.2. Fuzzy Oil Drop Model—FOD-M

This model has already been described [20]. The description given here is intended to facilitate the interpretation of the results presented.

The model is used to assess the distribution of hydrophobicity in the protein body. The actual distribution—O (*H^O^* in Equation (1))—resulting from inter-residual interactions is expressed in the model in question by a function proposed by Levitt [37].(1)$$H_{i}^{O}=\frac{1}{H_{sum}^{O}}\sum_{j}\left\{\begin{array}{l} \left(H_{i}^{r}+H_{j}^{r}\right)\left(1-\frac{1}{2}\left(7{\left(\frac{r_{ij}}{c}\right)}^{2}-9{\left(\frac{r_{ij}}{c}\right)}^{4}+5{\left(\frac{r_{ij}}{c}\right)}^{6}-{\left(\frac{r_{ij}}{c}\right)}^{8}\right)\right)\;for\;r_{ij}\leq c\\ 0,\;\;for\;r_{ij}>c \end{array}\right.$$
where *r_ij_* is the distance between the effective atoms (the averaged position of the atoms belonging to a given amino acid) and *C* is cutoff distance = 9 Ǻ (after [24]). *H^r^* are parameters expressing the intrinsic hydrophobicity of a given amino acid. Any scale of hydrophobicity can be applied [38].

This distribution is compared with the idealised distribution T (*H^T^* in Equation (2)), which assumes a micelle-like distribution. This distribution is achieved by encapsulating the protein molecule in an appropriate 3D Gaussian function.(2)$$H_{i}^{T}=\frac{1}{H_{sum}^{T}}\exp\left(\frac{-{\left(x_{i}-\bar{x}\right)}^{2}}{2\sigma_{x}^{2}}\right)\exp\left(\frac{-{\left(y_{i}-\bar{y}\right)}^{2}}{2\sigma_{y}^{2}}\right)\exp\left(\frac{-{\left(z_{i}-\bar{z}\right)}^{2}}{2\sigma_{z}^{2}}\right)$$

The values of the *σ_X_*, *σ_Y_*, and *σ_Z_* parameters are adapted to the size and shape of the protein molecule.

The values of the Ti function assigned to the positions of the effective atoms determine the micelle-like compatible, idealised hydrophobicity level. This assumption is based on an analysis of the phenomenon of micelle formation in aqueous environments by bipolar molecules. By treating amino acids as bipolar molecules with varying proportions of polar to hydrophobic parts, it is assumed that the chain being folded in an aqueous environment tends to form an arrangement with a hydrophobic centric core and a polar surface. Micelle-like ordering is achieved to varying degrees. The ordering degree of distribution O against distribution T (reference distribution) is determined using divergence entropy introduced by Kullback–Leibler [39].(3)$$D_{KL}(P|Q)=\sum_{i=1}^{N}P_{i}\log_{2}\frac{P_{i}}{Q_{i}}$$

In the FOD-M model, distribution *P* is the distribution analysed—in our model, it is distribution O. Distribution *Q* in Equation (3) is the reference distribution—in the FOD-M model, it is distribution T.

The value determined by Equation (3) cannot be interpreted. Therefore, a second reference distribution was introduced—R, in which *Ri* = *1*/*N*, where *N* is the number of amino acids in the chain. This distribution assigns an equal level of hydrophobicity to all amino acids. This distribution implies an even distribution of hydrophobicity across all amino acids, implying the absence of a hydrophobic core.

The value *D_KL_*(*O|T*) determines the distance of the O distribution to the T distribution, while *D_KL_*(*O|R*) is the distance of the O distribution to the R distribution. The relationship *D_KL_*(*O|T*) < *D_KL_*(*O|R*) indicates the proximity of the O distribution to the T distribution and is therefore interpreted as the presence of a hydrophobic core in the protein structure.

In order to avoid operating with two values describing the same object, the measure *RD*—relative distance—was introduced.(4)$$RD=\frac{D_{KL}(O|T)}{D_{KL}\left(O|T\right)+D_{KL}(O|R)}$$

An *RD* value < 0.5 indicates the presence of a hydrophobic core. The comparison of *T*, *O*, and *R* distribution is shown in Figure 16A. The presentation is reduced to 1D to make the presentation readable. The calculated *RD* value for the example shown in Figure 16A is presented in Figure 16B.

The aqueous environment is not the only one in which proteins exhibit biological activity. The biological membrane, which is the environment for many proteins, has different characteristics. A membrane protein for stabilisation in the membrane environment requires the exposure of hydrophobic residues for preferable contact with the membrane environment. The following function should be used to describe such a distribution:*M_i_* = *T_MAX_* − *T_i_*(5)

The distribution expected for the aqueous environment and for the hydrophobic environment of the membrane is shown in Figure 1B.

An analysis of numerous proteins identified a universal solution applicable to all proteins in the form of the following function:(6)$$M_{i}={\left[T_{i}+{K(T_{MAX}-T_{i})}_{n}\right]}_{n}\;$$

Function M expresses the contribution of both the 3D Gaussian field (polar environment) and the inverse of this function (hydrophobic environment). An important role is played by the value of the K parameter, which expresses the contribution of non-aqueous factors to the construction of the external force field for polypeptide chain folding.

The value of K = 0 indicates the exclusive contribution of the aqueous environment, while high values of the K parameter indicate a significant contribution of non-aqueous factors including hydrophobic factors in particular.

Proteins with K = 0 are down-hill, fast-folding, and ultra-fast-folding proteins [40]. Membrane proteins show K-parameter values in the range 0.9–3.5 [41].

A graphical representation of the model used, shown in Figure 16, illustrates the steps in the procedure for determining the values of the RD and K parameters characterising the protein in question.

Determining the value of parameter K is carried out in an iterative procedure in search of the minimum value of D_KL_(O|M) (Figure 16C). The recognition of K value for the set of profiles T, O, and M can be defined for particular examples (Figure 16A: grey line).

### 4.3. Programs Used

The program allowing calculation of RD as well as T and O distribution is accessible upon request on CodeOcean platform: https://codeocean.com/capsule/3084411/tree, accessed on 27 June 2025. Please contact the corresponding author to get access to your private program instance.

The application is implemented in collaboration with the Sano Centre for Computational Medicine (https://sano.science, accessed on 27 June 2025) and runs on resources contributed by ACC Cyfronet AGH (https://www.cyfronet.pl, accessed on 27 June 2025). The resources in the framework of the PL-Grid Infrastructure (https://plgrid.pl, accessed on 27 June 2025) provide a web wrapper for the abovementioned computational component and are freely available at https://hphob.sano.science, accessed on 27 June 2025.

The VMD program was used to present the 3D structures [25,42].

## 5. Conclusions

The results presented herein discuss examples of the use of the environment in protein folding—protein structuralisation. The roles of chaperonin Hsp104 and disaggregase as external force field providers for polypeptide chain folding have been demonstrated. The role of this field is to eliminate conditions resulting from the specific polar aqueous environment in favour of a field whose status is determined by the high value of the K parameter and the approximation of the hydrophobicity distribution to a rectilinear distribution (R distribution). The generated polypeptide chain structure is not necessarily the final structural product. It can provide a so-called starting structural form for the folding process in aqueous environments after leaving the chaperonin environment. The choice of a starting structure is critical for numerical structure prediction techniques operating a procedure to find the minimum energy of the structure of a given protein. The effectiveness of the internal force field optimisation procedure is highly dependent on the starting form of the polypeptide chain. This phenomenon was also shown in relation to dynamic forms related to the function of complexing the polypeptide chain by Hsp104. The role of the appropriate external force field is also revealed in the analysis of incorrectly folded proteins, which was demonstrated by referring to the set of misfolded proteins, the structures of which are available within the CASP project, in particular in CASP16 for the T1266_1-D1 target.

## Figures and Tables

**Figure 1 ijms-26-06360-f001:**
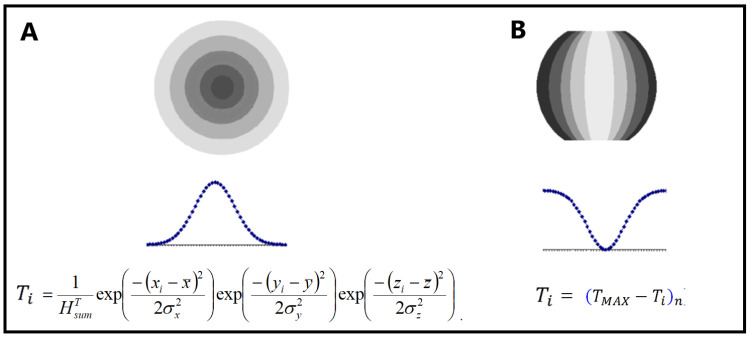
Gaussian function for the description of the idealised hydrophobicity distribution for. (**A**) aqueous environment. (**B**) hydrophobic membrane environment. The more dark color the higher hydrophobicity.

**Figure 2 ijms-26-06360-f002:**
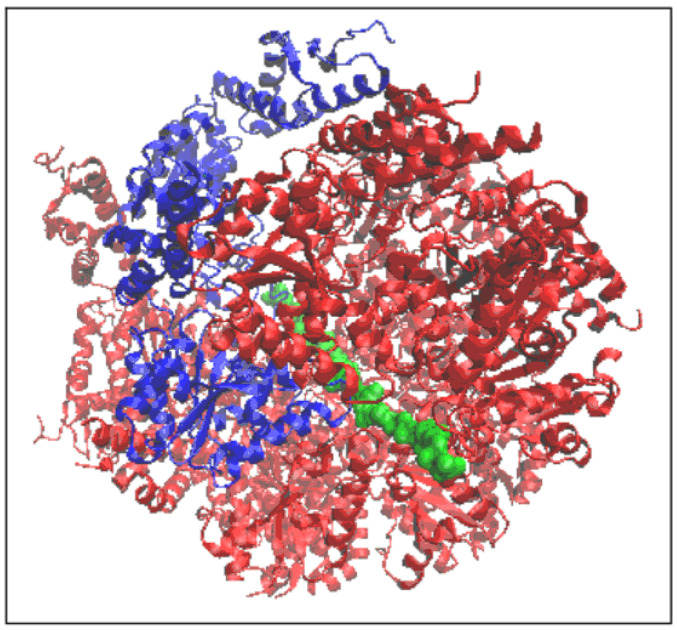
3D presentation of the structure forming the object of analysis 3D structure in red. In the arrangement of the six chaperonin-building chains, chain A (blue) is distinguished. The casein chain being folded in the chaperonin is marked as green.

**Figure 3 ijms-26-06360-f003:**
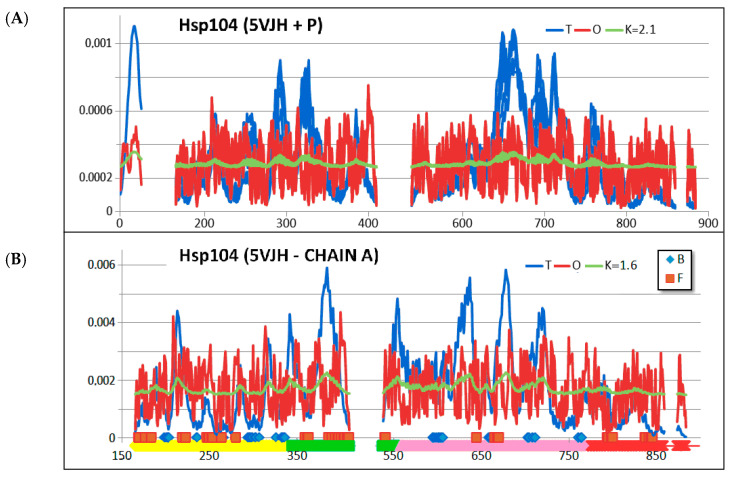
Overview of the T, O and M profiles for the given K value. (**A**) complete complex—casein section highlighted (red) on the horizontal axis—3D Gaussian function generated for the complete complex. (**B**) summary of the T, O, and M profiles for chain A treated as an individual structural unit. A 3D Gaussian function was generated for chain A only. The positions of residues involved in interactions with other chaperonin chains are highlighted on the horizontal axis, with chain B—blue and chain F—red. Horizontal axis bottom—division into pseudo-domains—colour scheme as in Figure 6.

**Figure 4 ijms-26-06360-f004:**
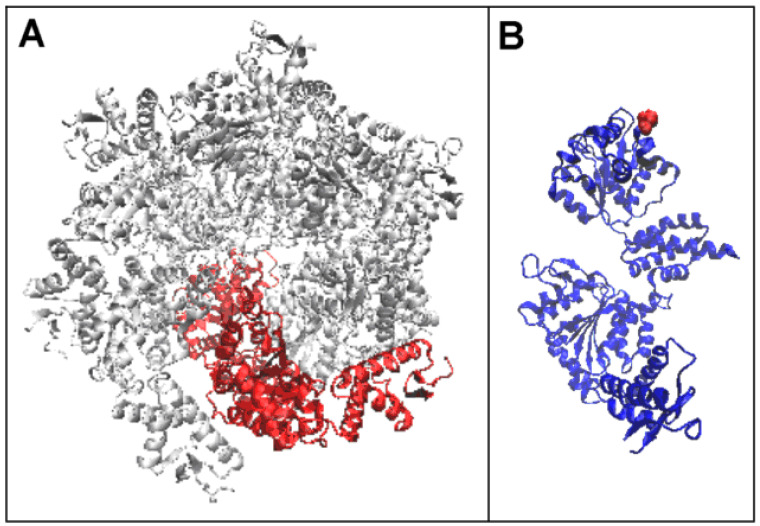
Structure of the single chain—Hsp104 component. (**A**) 3D presentation of the chain (red) as a component of the complex (white). (**B**) 3D presentation of a single chain—N-terminal position—red.

**Figure 5 ijms-26-06360-f005:**
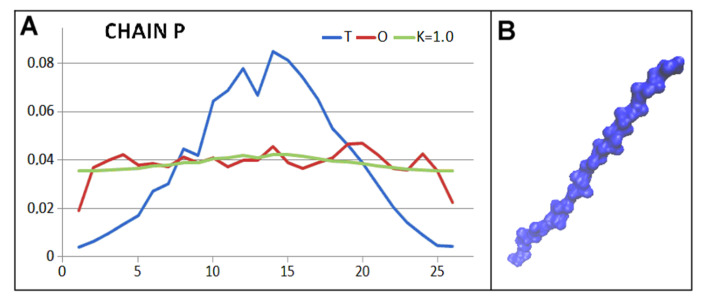
Characteristics of casein. (**A**) profiles T, O, and M (for the K value given in the title) for the P chain (casein) folded in the Hsp104 environment. (**B**) 3D presentation of casein as it appears in the complex with Hsp104.

**Figure 6 ijms-26-06360-f006:**
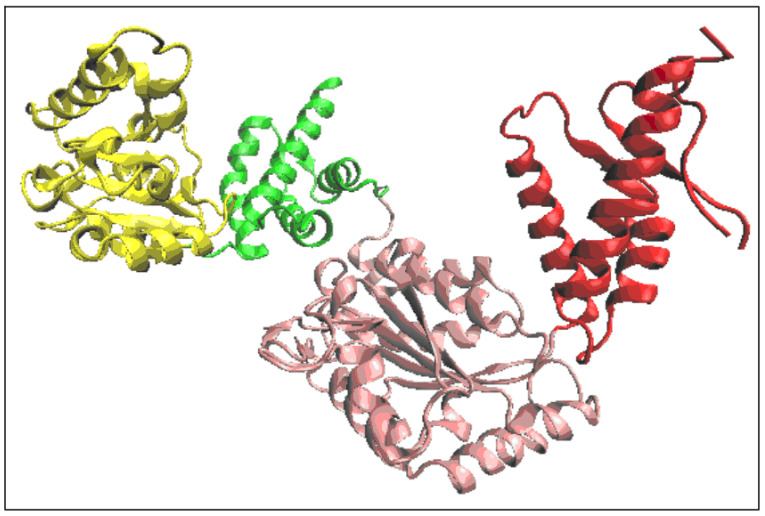
The proposed division of the A chain into subunits—pseudo-domains. The ranges of the chain fragments are given in Table 1.

**Figure 7 ijms-26-06360-f007:**
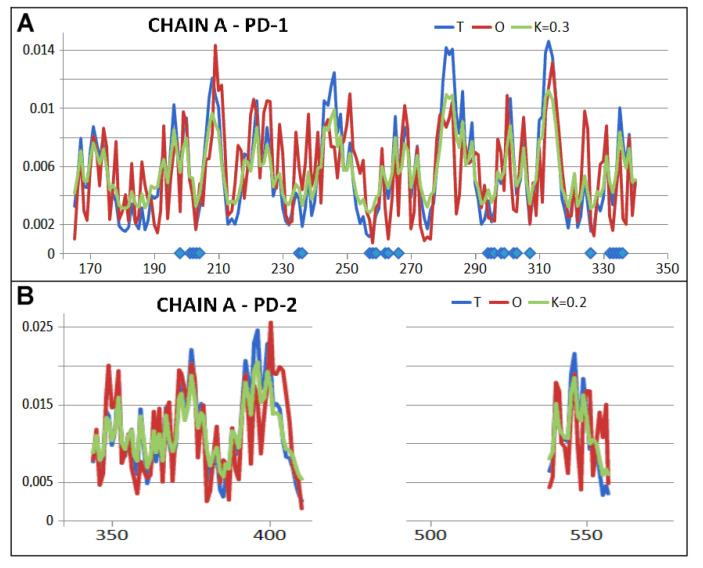
Status of pseudo-domains (PS) present in the single chain structure—the Hsp104 chaperonin component. (**A**) PD1 profiles T, O, and M (for the K value given in the title). The horizontal axis indicates the positions of the residues involved in the interaction with another chain. (**B**) T, O, and M profiles (for the K value given in caption).

**Figure 8 ijms-26-06360-f008:**
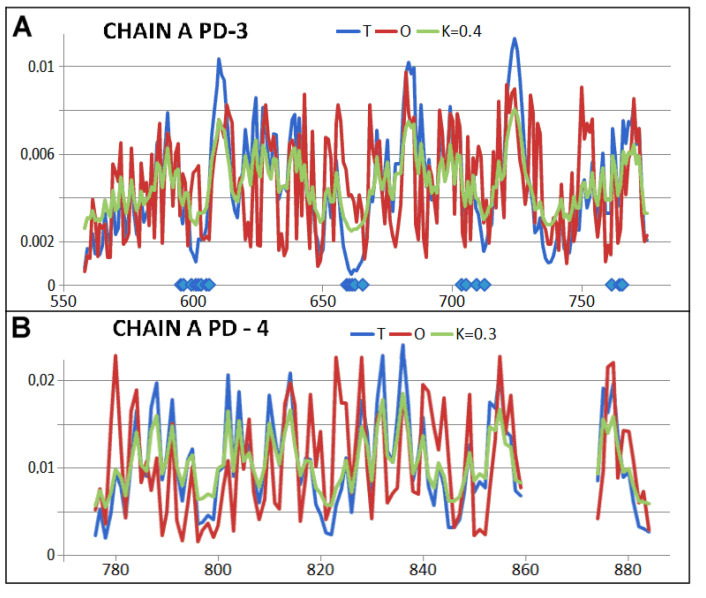
Status of pseudo-domains (PS) present in the single chain structure—the Hsp104 chaperonin component. (**A**) profiles T, O, and M (for the K value given in the caption)—the horizontal axis indicates the positions involved in the interchain interaction. (**B**) profiles T, O, and M (for the K value given in the caption—top).

**Figure 9 ijms-26-06360-f009:**
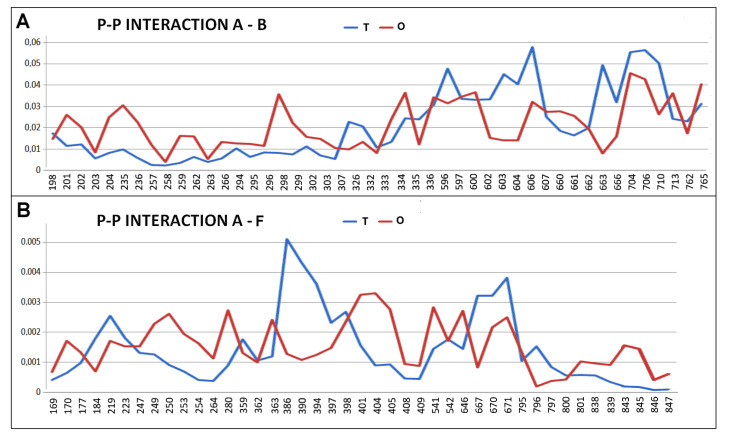
Status of A-chain amino acids involved in interchain interactions. (**A**) chain A in an interaction with chain B—numbers represent the positions in chain A. (**B**) chain A in an interaction with chain F—numbers represent the positions in chain A.

**Figure 10 ijms-26-06360-f010:**
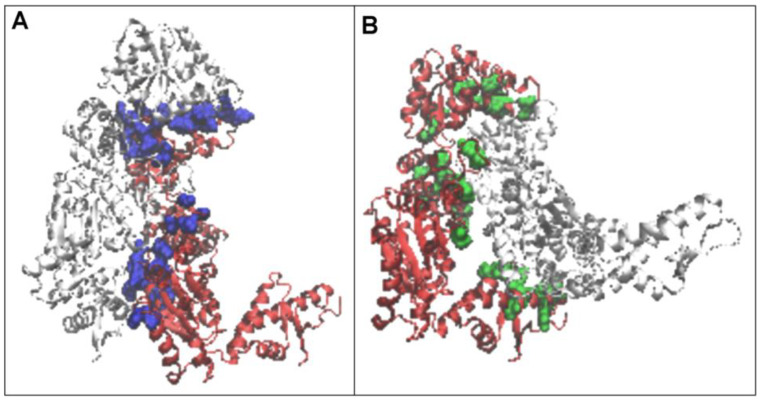
Interaction of chain A (red) with chain B (gray) and chain F (gray). (**A**) interaction with chain B (blue). (**B**) interaction with chain F (green).

**Figure 11 ijms-26-06360-f011:**
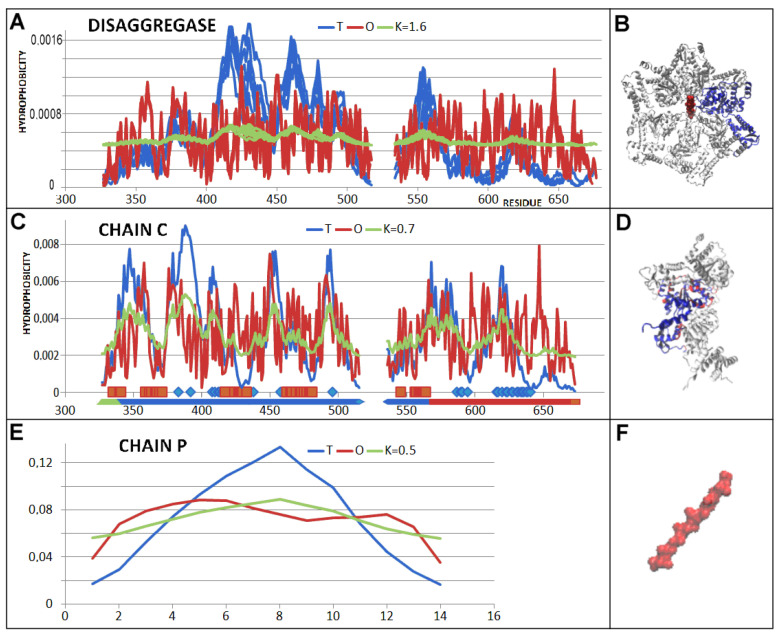
Summary of the T, O, and M profiles for the K value given in the legend expressing the disaggregase status (PDB ID—7TTR). (**A**) chains A–F—all chains overlapped. (**B**) 3D presentation of disaggregate with chain A distinguished in blue and chain P—casein in red. (**C**) chain C with residues engaged in PP interaction as shown on horizontal axis (upper): with chain A—red, and chain D—blue, bottom—colours distinguish the pseudo-domains as shown in Table 2. (**D**) chain C with 3D presentation—chain C—blue, residues interacting with chains B and D distinguished by red. Chains B and D—white. (**E**) profiles: T, O, and M for K given in legend of chain P—casein. (**F**) 3D presentation of chain P—casein.

**Figure 12 ijms-26-06360-f012:**
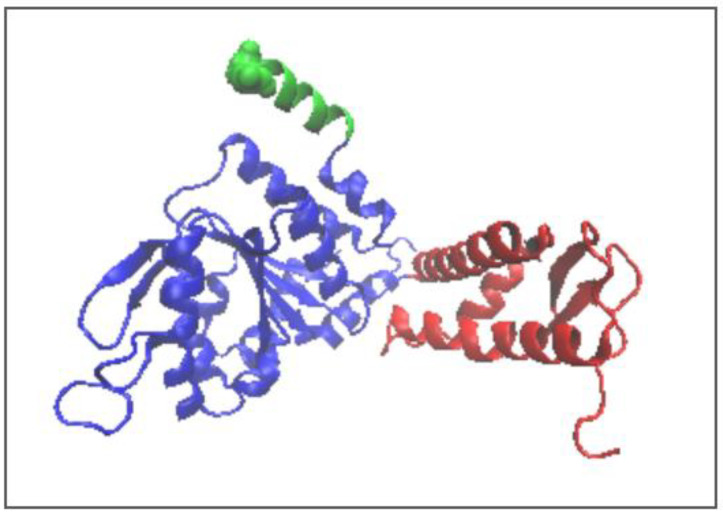
3D presentation of chain A of disaggregase. Colours define the pseudo-domains as shown in Figure 11C on bottom horizontal line.

**Figure 13 ijms-26-06360-f013:**
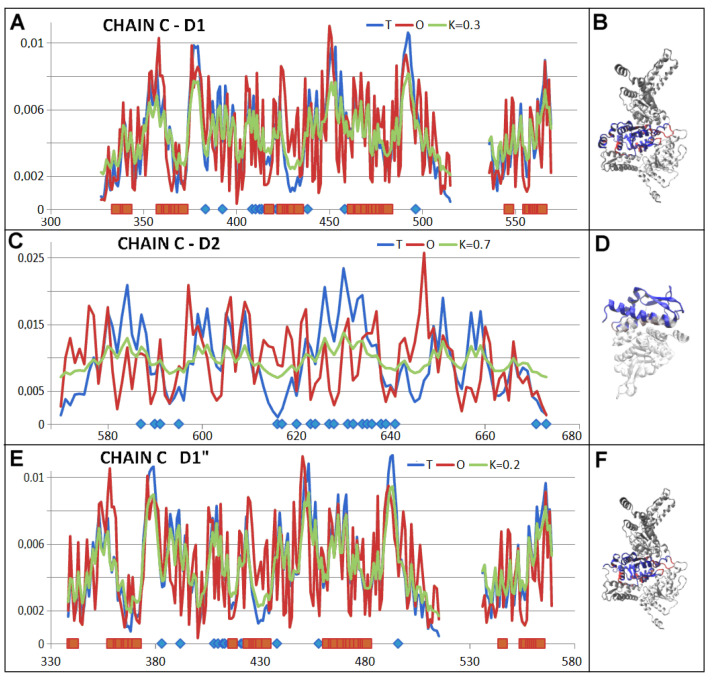
Summary of the T, O, and M profiles for the K value given in the legend. (**A**) chain C—PD1, residues engaged in P-P interactions: with chain A—red, and with chain D—blue. (**B**) 3D presentation of the structure of this domain with highlighted sections involved in an interaction with other chains—colours as given in A. (**C**) chain C—PD2—on horizontal axis—interaction with chain D. (**D**) 3D presentation of domain 2 with highlighted fragments involved in an interaction with other chains of the complex—as given in C. (**E**) chain C domain 1—another form of PD1 domain lacking a loose helical segment (327–326). The N-terminal fragment (green on Figure 12) eliminated as representing relatively low packing with the rest of the PD1. (**F**) 3D presentation of the domains given in blue with residues engaged in interactions—red. Chains adjacent to C—chains B and D given in white.

**Figure 14 ijms-26-06360-f014:**
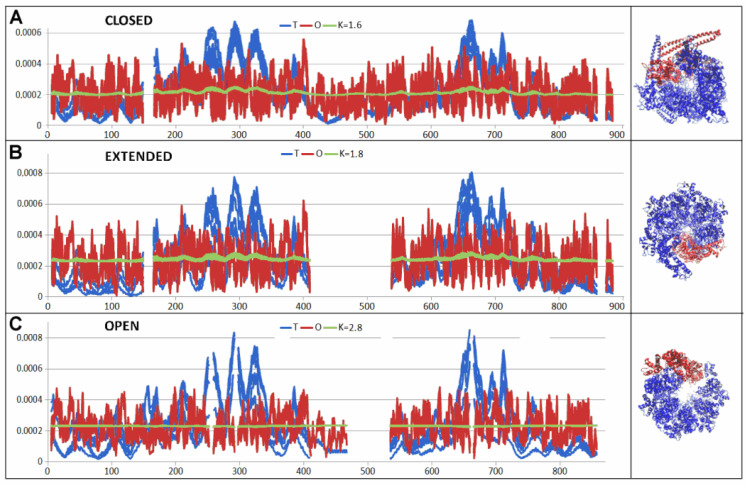
Profiles T, O, and M for K given in legend. (**A**) all chains in closed form. (**B**) all chains in extended form. (**C**) all chains in open form. The 3D presentation for each example is in right column. Red chain—chain F.

**Figure 15 ijms-26-06360-f015:**
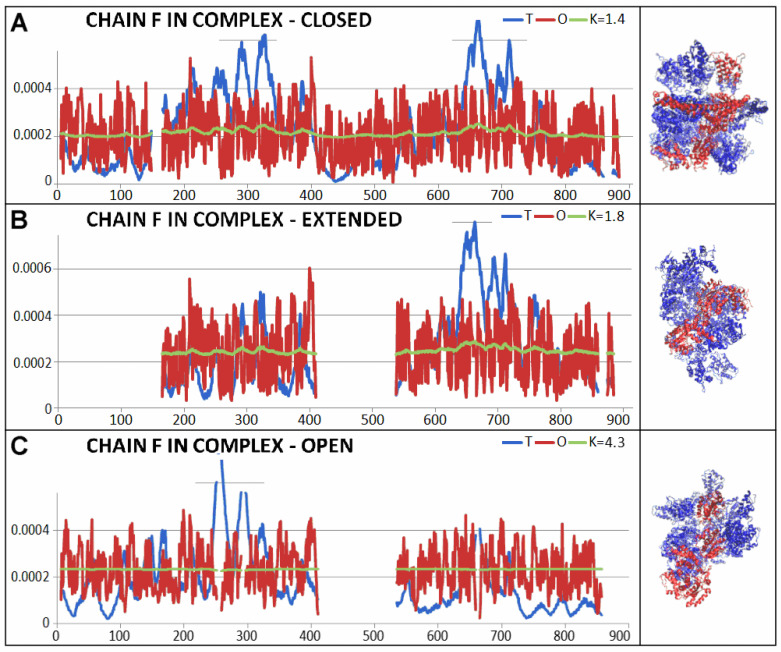
Profiles T, O, and M for K given in legend representing F chain as part of complex. (**A**) closed form. (**B**) extended form. (**C**) open form. The 3D presentation for each example in right column. Red chain—chain F.

**Figure 16 ijms-26-06360-f016:**
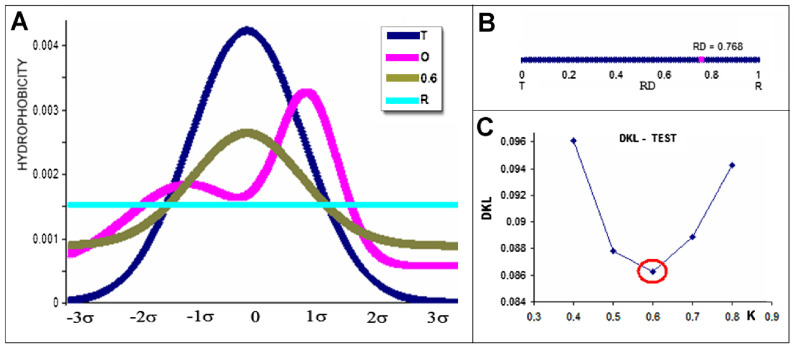
Steps in applying the procedure based on the FOD-M model—reduced to 1D. (**A**) profiles: T (blue), O (pink), R (turquoise), M for K = 0.6—grey. (**B**) RD value determined for the relationship of distribution O for the reference distributions T and R (as shown in A). (**C**) K value determination as the value showing the minimum for D_KL_(O|M).

**Table 1 ijms-26-06360-t001:** Summary of parameter values for the structural units highlighted in the analysis.

5VJH	Fragment	Individual		In Complex	
		RD	K	RD	K
	CHAINS A-F + P	0.729	2.1		
	CHAINS A-F	0.729	2.1		
CHAIN A		0.722	1.6	0.724	
CHAIN B		0.702	1.3	0.732	
CHAIN C		0.668	1.1	0.709	
CHAIN D		0.722	1.1	0.710	
CHAIN E		0.734	1.1	0.747	
CHAIN F		0.755	1.5	0.737	
CHAIN P (casein)		0.955	1.0	0.781	
PSEUDO-DOMAINS	165–340	0.409	0.3		
	344–557	0.367	0.2		
	558–775	0.490	0.4		
	776–884	0.459	0.3		

**Table 2 ijms-26-06360-t002:** Summary of parameter values determining the status of the listed structural units in disaggregase (PDB ID—7TTR): chains treated as parts of complex, and chains treated as individual structural units. Pseudo-domains (not identified in the PDBSum database) are highlighted. The 3D presentation of pseudo-domains is shown in Figure 11.

		Chain in Complex		Individual	
Chain	Fragment	RD	K	RD	K
A-F				0.743	1.6
A-F + P				0.740	
A				0.609	0.6
B				0.637	0.7
C		0.737	1.6	0.613	0.7
D		0.758	1.6	0.591	0.5
E		0.749	1.3	0.544	0.4
F		0.675	1.2	0.557	0.5
P		0.745	1.7	0.802	0.5
C—PD 1	(327–569)	0.758	1.7	0.577	0.4
C—PD 2	(570–653)	0.751	0.5	0.624	0.6
C—PD 3	(338–569)			0.556	0.3
Chain P-P				0.786	
Chain No P-P				0.655	

**Table 3 ijms-26-06360-t003:** The values of the RD and K parameters for complexes and chains treated as complex components and for the chains treated as individual structural units.

PDB ID	Complex		Chain in Complex		Chain Individual	
Hsp104dwb	RD	K	RD	K	RD	K
6N8T—closed conformation	0.702	1.6				
Chains: A			0.711	1.6	0.748	1.4
B			0.683	1.4	0.748	1.4
C			0.705	1.5	0.739	1.4
D			0.689	1.5	0.776	1.6
E			0.716	1.7	0.744	1.4
F			0.706	1.4	0.735	1.2
6N8V—open conformation	0.784	2.8				
Chains: A—704 AA			0.800	2.5	0.808	1.8
B—704 AA			0.800	2.2	0.845	2.1
C—704 AA			0.741	2.0	0.813	2.0
D—769 AA			0.769	2.2	0.782	1.6
E—704 AA			0.789	2.7	0.826	2.1
F—704 AA			0.786	4.3	0.824	1.2
6N8Z—extended conformation	0.734	1.8				
Chains: A—579 AA			0.740	1.7	0.701	1.3
B—723 AA			0.738	1.8	0.760	1.5
C—723 AA			0.702	1.6	0.763	1.6
D—723 AA			0.751	1.7	0.737	1.4
E—723 AA			0.760	1.8	0.742	1.4
F—579 AA			0.692	1.7	0.703	1.2

## Data Availability

The potential user has two possible ways to access the program: The program allowing the calculation of RD as well as T and O distribution is accessible upon request on the CodeOcean platform: https://codeocean.com/capsule/3084411/tree, accessed on 27 June 2025. Please contact the corresponding author to get access to your private program example. The application—implemented in collaboration with the Sano Centre for Computational Medicine (https://sano.science, accessed on 27 June 2025) and running on resources contributed by ACC Cyfronet AGH (https://www.cyfronet.pl, accessed on 27 June 2025) in the framework of the PL-Grid Infrastructure (https://plgrid.pl, accessed on 27 June 2025)—provides a web wrapper for the abovementioned computational component and is freely available at https://hphob.sano.science, accessed on 27 June 2025.

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
