# Peer review of "Heat Shock Protein and Disaggregase Influencing the Casein Structuralisation"

_ijms, 2025, doi:10.3390/ijms26136360_

Round 1

Reviewer 1 Report

Comments and Suggestions for Authors

The manuscript titled “HEAT-SHOCK PROTEIN AND DISAGGREGASE INFLU- 2 ENCING THE CASEIN STRUCTURALISATION” aimed to discusse an example of the action of the Hsp104 chaperonin as a protein that assists in the process of polypeptide chain folding by providing an external force field to which the polypeptide chain being folded adapts. An analysis of the role of the analogous proteins supporting the folding process - prefoldin , chaperone and chaperonin indicates a similar mechanism for the folding process in the presence of chaperone proteins. The manuscript is more like a preliminary research work and the data of the manuscript could not support the conclusion well, so that the manuscript is not suitable for publication currently.

Although the authors used relatively novel analytical methods, all analyses were based on static structures and did not reflect the dynamic interactions between chaperone and chaperonin.

The authors need to try to compare the differences between the correct folded structure and the incorrectly folded structure to further prove the accuracy of the conclusion.

The author needs to count more proteins, not just these few proteins in the manuscript, to draw corresponding conclusions.

Should the author attempt to analyze the relationship between amino acid distribution and the folding driving forces found?

The resolution of all structural figuresin the manuscript should be increased

Author Response

REVIEWER 1

Dear Reviewer 1

Many thanks for comments. We did our best to follow yor suggestions.

All newly introduced fragments are given in red as well as complete Supplementary Materials.

We hope, the final form appears acceptable for publications.

Sincerely yours;

Irena Roterman                                                                                       Krakow, June 16, 2025

Comments and Suggestions for Authors

The manuscript titled “HEAT-SHOCK PROTEIN AND DISAGGREGASE INFLU- 2 ENCING THE CASEIN STRUCTURALISATION” aimed to discusse an example of the action of the Hsp104 chaperonin as a protein that assists in the process of polypeptide chain folding by providing an external force field to which the polypeptide chain being folded adapts. An analysis of the role of the analogous proteins supporting the folding process - prefoldin , chaperone and chaperonin indicates a similar mechanism for the folding process in the presence of chaperone proteins. The manuscript is more like a preliminary research work and the data of the manuscript could not support the conclusion well, so that the manuscript is not suitable for publication currently.

Although the authors used relatively novel analytical methods, all analyses were based on static structures and did not reflect the dynamic interactions between chaperone and chaperonin.

IR – The additional structures represent the dynamic forms related to the function. This analysis allows identification of chains-parts of complex responsible for function. The analysis of their structures reveals the necessary structural changes.

The authors need to try to compare the differences between the correct folded structure and the incorrectly folded structure to further prove the accuracy of the conclusion.

IR – The set of misfolded proteins is analysed and described in supplementary Materials. The collection of models for CASP16 target T1266_1-D1 has been selected. This relatively large seto of examples reveals the role of RD and K parameters. The discussed proteins do not represente the Hsp group. This is why they are presented separately in Supplementary Materials.

The author needs to count more proteins, not just these few proteins in the manuscript, to draw corresponding conclusions.

IR – the new three examples are supplemented and analysed using FOD-M model.

Should the author attempt to analyze the relationship between amino acid distribution and the folding driving forces found?

IR – This is the interesting issue however it requires independent examination beyond the subject of the presented problem.

The resolution of all structural figures in the manuscript should be increased

IR – I am sorry – This is the best we can do.

Reviewer 2 Report

Comments and Suggestions for Authors

This manuscript reports the original discovery of characterizing protein folding and refolding effects of a heat shock protein and disaggregase HSP104 to casein using a previously described model FOD-M. The proposed model uses the values of K parameter to describe the environmental variability or external force field in the folding process, and RD parameter to describe the hydrophobic distribution of the folded protein. It is concluded that external force field is important for protein folding. This manuscript can be further improved to be considered for acceptance.     

Comments:

#1: It is recommended to include one or more additional examples of complexes of heat shock protein and disaggregase (e.g. ClpB) and casein to better support the discussions and/or conclusions, even if the information of additional examples may have been previously described.

#2: Inconsistent font sizes or types in the text and figure captions should be corrected, and high-resolution images should be provided for several figures.

#3: References are missing in the second and third paragraphs of the Introduction.

#4: In line 169, it is unknown where the Fig. 1D is.

Author Response

REVIEWER 2

Dear Reviewer 2

Many thanks for comments. We did our best to follow yor suggestions.

All newly introduced fragments are given in red as well as complete Supplementary

We hope, the final form appears acceptable for publications.

Sincerely yours;

Irena Roterman                                                                                       Krakow, June 16, 2025

Comments and Suggestions for Authors

This manuscript reports the original discovery of characterizing protein folding and refolding effects of a heat shock protein and disaggregase HSP104 to casein using a previously described model FOD-M. The proposed model uses the values of K parameter to describe the environmental variability or external force field in the folding process, and RD parameter to describe the hydrophobic distribution of the folded protein. It is concluded that external force field is important for protein folding. This manuscript can be further improved to be considered for acceptance.     

Comments:

#1: It is recommended to include one or more additional examples of complexes of heat shock protein and disaggregase (e.g. ClpB) and casein to better support the discussions and/or conclusions, even if the information of additional examples may have been previously described.

IR – The set of proteins examined is bigger now. Three additional disaggregases in their different function-related structural forms are discussed in the current form of the paper.

References to another papers discussing the other chaperons/chaperonins role in folding process are present on the list of references [17,18,19].

#2: Inconsistent font sizes or types in the text and figure captions should be corrected, and high-resolution images should be provided for several figures.

IR – Font size corrected.

#3: References are missing in the second and third paragraphs of the Introduction.

IR – References supplemented.

#4: In line 169, it is unknown where the Fig. 1D is.

 IR – Corrected.

Round 2

Reviewer 1 Report

Comments and Suggestions for Authors

The authors have addressed most of my concerns, but the resolution of the Figures needs to be further improved before formal publication. In addition, it is recommended that the authors use molecular dynamics simulations to further analyze the dynamic interactions between molecular chaperones and chaperones.

Author Response

Dear Reviewer

Many thanks for comments

Ad.1. - Simulation of molecular dynamics - this is not any object of our analysis. The status of hydrophobicity distribution in chaperonine is treated as external force field in which the folding proteins stabilises its expected structure. Molecular dynamics simulation is a very good tool for other type of analysis.

However many thanks for this suggestion. We plan to use this tool in close future for structural analysis. The newly supplemented three structures represent dynamic forms related to biological activity. Thisi swhy it is treated as dynamic forms of the chaperonin. 

Ad. 2. The quality of figures. 

We are sorry - this is the best we can do in this matter. 

Sincerely yours;

Irena Roterman  

Reviewer 2 Report

Comments and Suggestions for Authors

This revised manuscript has addressed comments put forth in previous review. The manuscript is accepted for publication as is.

Author Response

Dear Reviewer

Many thanks for your suggestions presented in former revision. 

Many thanks for accepting our paper for publcation. 

Sincerely yours;

Irena Roterman